TEffectR: an R package for studying the potential effects of transposable elements on gene expression with linear regression model

Karakülah Gökhan gokhan.karakulah@deu.edu.tr 1 2
Arslan Nazmiye 1
Yandım Cihangir 1 3
Suner Aslı asli.suner@ege.edu.tr 4
1 Izmir Biomedicine and Genome Center , Izmir , Turkey
2 Izmir International Biomedicine and Genome Institute, Dokuz Eylül University , Izmir , Turkey
3 Department of Genetics and Bioengineering, Faculty of Engineering, Izmir University of Economics , Izmir , Turkey
4 Department of Biostatistics and Medical Informatics, Faculty of Medicine, Ege University , Izmir , Turkey
Papaleo Elena
Electronic publication date: 2019 Dec 5
Publication date: 2019
Volume: 7
Electronic Location ID: e8192
Received 2019 Aug 15; Accepted 2019 Nov 11
Copyright: ©2019 Karakülah et al.
Copyright year: 2019
Copyright holder: Karakülah et al.
License: This is an open access article distributed under the terms of the Creative Commons Attribution License, which permits unrestricted use, distribution, reproduction and adaptation in any medium and for any purpose provided that it is properly attributed. For attribution, the original author(s), title, publication source (PeerJ) and either DOI or URL of the article must be cited.
License URL: https://creativecommons.org/licenses/by/4.0/

Keywords: Transposable elements, Gene regulation, Gene expression, Regression, Linear model, R package

Funding: The authors received no funding for this work.

==============================
Introduction

Recent studies highlight the crucial regulatory roles of transposable elements (TEs) on proximal gene expression in distinct biological contexts such as disease and development. However, computational tools extracting potential TE –proximal gene expression associations from RNA-sequencing data are still missing.

Implementation

Herein, we developed a novel R package, using a linear regression model, for studying the potential influence of TE species on proximal gene expression from a given RNA-sequencing data set. Our R package, namely TEffectR, makes use of publicly available RepeatMasker TE and Ensembl gene annotations as well as several functions of other R-packages. It calculates total read counts of TEs from sorted and indexed genome aligned BAM files provided by the user, and determines statistically significant relations between TE expression and the transcription of nearby genes under diverse biological conditions.

Availability

TEffectR is freely available at https://github.com/karakulahg/TEffectR along with a handy tutorial as exemplified by the analysis of RNA-sequencing data including normal and tumour tissue specimens obtained from breast cancer patients.

Introduction

Transposable elements (TEs) are DNA sequences that are able to translocate themselves along a host genome (Biemont & Vieira, 2006). They were discovered by Barbara McClintock in the 1950s in maize and defined for the first time as controlling elements on the action of nearby genes (McClintock, 1956). TEs constitute a considerable portion of most eukaryotic genomes (Kazazian Jr, 2004; Kelly & Leitch, 2011; Lander et al., 2001) and are divided into two main classes according to their transposition mechanism (Wicker et al., 2007). Class I elements (also known as retrotransposons) use RNA intermediates and a reverse transcriptase whereas Class II elements (also known as DNA transposons) act through DNA intermediates for their translocation (Wicker et al., 2007). These mechanisms are also called as “copy-and-paste” and “cut-and-paste” transpositions, respectively. In addition to acting as key players in genome size expansion and evolution (Kazazian Jr, 2004), previous studies highlighted the critical roles of TEs in distinct biological contexts, such as cancer (Hancks & Kazazian Jr, 2016; Johanning et al., 2017; Lee et al., 2012), embryonic development (Yandim & Karakulah, 2019b), senescence and aging (De Cecco et al., 2019), and stress response (Rech et al., 2019).

In parallel to the advent of next generation sequencing technologies, considerable attention has been paid to elucidate the regulatory activities of TEs on gene expression on a genome-wide scale. TEs are now recognized as the natural source of diverse regulatory sequences (Trizzino et al., 2017) including the promoters (Jordan et al., 2003), transcription factor binding sites (Bourque et al., 2008; Karakulah, 2018), enhancers (Chuong et al., 2013), and silencers (Bire et al., 2016) in the host genome. For example, MER39, a human long terminal repeat (LTR), acts as an endometrium-specific promoter and plays an essential role for the expression of the prolactin gene during pregnancy (Emera et al., 2012). Similarly, an MT-C retrotransposon-derived promoter is required to produce the oocyte-specific isoform of the Dicer gene in mice and its absence leads to female infertility (Flemr et al., 2013). It has also been reported in a comprehensive computational study that the majority of primate-specific regulatory sequences are originated from TEs (Jacques, Jeyakani & Bourque, 2013). In line with this, the influence of TEs on proximal gene expression was documented both in rat (Dong et al., 2017) and maize (Makarevitch et al., 2015). Furthermore, housekeeping genes were distinguished by their distinct repetitive DNA sequence environment (Eller et al., 2007). When it comes to understanding the links between TEs and proximal genes, it is postulated that TE intermediates (DNA or RNA) may interfere with the transcription of adjacent genes either directly or through recruited factors, and that an activated or repressed TE has the potential to modulate the chromatin environment of such genes and thereby influence their expression states (Elbarbary, Lucas & Maquat, 2016; Huda et al., 2009).

Despite the above-mentioned efforts on dissecting the influence of TEs on the expressions of proximal genes, a systematic and statistically valid approach is still missing, particularly due to the fact that TEs have many copies in the genome. In other words, it is challenging to link a particular TE in a specific location to a particular gene of interest. Still, a notable effort has been devoted to developing computational methods on the matter. Among these, two online tools, PlanTEnrichment (Karakulah & Suner, 2017) and GREAM (Chandrashekar, Dey & Acharya, 2015), allow their users to determine overrepresented TEs that are located adjacently of a given list of genes in plants and mammals, respectively. RTFAdb (Karakulah, 2018), using transcription factor binding profiles of the Encyclopedia of DNA Elements (ENCODE) project (The ENCODE Project Consortium’, 2012), can be utilized for exploring the regulatory roles of TEs. TETools (Lerat et al., 2017) and RepEnrich (Criscione et al., 2014) are popular computational tools to study differential expression of TEs under different biological conditions. Additionally, RepEnrich can help to provide insights into the transcriptional regulation of TEs by linking chromatin immunoprecipitation followed by sequencing (ChIP-seq) and expression profiling data sets. However, these tools do not allow one to directly link the expression of location specific TEs to a given proximal gene. Hence, we developed a novel R (https://www.r-project.org) package, using linear regression model (LM), for dissecting significant associations between TEs and proximal genes in a given RNA-sequencing (RNA-seq) data set. Our R package, namely TEffectR, makes use of publicly available RepeatMasker TE (http://www.repeatmasker.org) and Ensembl gene annotations (https://www.ensembl.org/index.html) and calculate total read counts of TEs from sorted and indexed genome aligned BAM files. Then, it predicts the influence of TE expression on the transcription of proximal genes under diverse biological conditions. In order to demonstrate the utility of TEffectR, we examined a publicly available RNA-seq data set collected from breast cancer patients. A detailed background of LM is also given in the following section.

Materials and Methods

Modeling gene expression with linear regression model

RNA-seq method yields count-type data rather than continuous measures of gene expression. Hence, generalized linear models (GLM) are used for modeling and statistical analysis of RNA-seq data sets, which are assumed to follow Poisson distribution or negative binomial distribution. In order to test differential gene expression, a number of analytical methods, including edgeR (Robinson, McCarthy & Smyth, 2010), and DESeq2 (Oshlack, Robinson & Young, 2010) use GLM where expression level of each gene is modeled as response variable while biological conditions (e.g., control vs experimental groups) are considered as explanatory variables or predictors. However, after the transformation of RNA-seq count data to log2-counts per million (logCPM) with Limma’s voom (Law et al., 2014) function, gene expression profiles can be ready for linear modelling.

LM has been used so far for modeling the regulatory effects of genetic and epigenetic factors on gene expression (Gerstung et al., 2015; Li, Liang & Zhang, 2014). For example, Li et al. developed RACER (Li, Liang & Zhang, 2014), using regression analysis approach, for exploring potential links between gene expression levels and a number of predictors, including DNA methylation level, copy number variation, transcription factor occupancy and microRNA expression level. Using a similar approach, TEffectR considers given biological conditions or covariates and TE expression levels as predictors to explain significant differences in gene abundances on a gene-by-gene basis. TEffectR assumes that each TE has a potential to influence the expression of a proximal gene. Accordingly, the expression level of a given gene can be modelled as follows: Geneexpressioni=β0+β1TE1i+⋯+βnTEni+βmCovaritesmi+εi,εi∼N0,σ2

• where i denotes the genes, and Geneexpressioni represents the normalized log2(CPM) value of the ith gene.

• TEni stands for the normalized log2(CPM) expression value of n th TE which is located within the upstream of the ith gene.

• Covaritesmi indicates covariate effects in the model, such as tissue type, age, gender, etc.

Implementation of the TEffectR package

The TEffectR package was written in R language (v.3.5.3) and it uses the functions of diverse computational tools. To extract gene annotation data from Ensembl database and manipulation of RepeatMasker annotation files, TEffectR respectively utilizes biomaRt (Durinck et al., 2009) and biomartr (Drost & Paszkowski, 2017) packages. The GenomicRanges (Lawrence et al., 2013) tool is used to identify TE sequences that are located in the neighborhood of the gene list provided by users. For data and string manipulation steps of the TEffectR workflow, we made use of dplyr (https://dplyr.tidyverse.org/), rlist (https://renkun-ken.github.io/rlist/) and stringr (https://stringr.tidyverse.org/) packages. BEDtools (Quinlan, 2014) and Rsamtools (https://bioconductor.org/packages/release/bioc/html/Rsamtools.html) were employed for the quantification of TE-derived sequencing reads in a given list of indexed and genome aligned BAM files. Two popular differential gene expression analysis packages for RNA-seq data sets, edgeR and limma, were used for filtering, normalization and transformation of expression values of both genes and TEs. Statistical significance of each LM and covariate effect in the corresponding regression model were calculated with lm(), which is a built-in function of R.

Data collection and processing for case study

In order to demonstrate the usage of TEffectR package, we made use of a publicly available whole transcriptome sequencing dataset including normal and tumour tissue specimens obtained from 22 ER+/HER2- breast cancer patients (GEO Accession ID: GSE103001) (Wenric et al., 2017). These transcriptome libraries were particularly included as they were prepared without poly(A) selection method and thereby allowing the measurement of TE expression uniformly (Solovyov et al., 2018). We downloaded sequencing reads in FASTQ file format from Sequence Read Archive (Leinonen et al., 2011) (SRA Accession ID: SRP116023) using SRA Tool Kit v.2.9.0 with “fastq-dump –gzip –skip-technical –readids –dumpbase –clip –split-3” command. Next, sequencing reads were aligned to the human reference genome GRCh38 (Ensembl version 78) using the splice-aware aligner HISAT2 v2.1.0 (Kim, Langmead & Salzberg, 2015) with “hisat2 -p -dta -x {input.index} -1 {input_1.fq} -2 {input_2.fq} -S {out.sam}” parameters. Stringtie v1.3.5 (Pertea et al., 2015) were used with “stringtie -e -B -p -G {input.gtf} -A {output.tab} -o {output.gtf} -l {input.label}{input.bam}” parameters for expression quantification at the gene level. We considered only the uniquely mapped reads overlapping TE regions for the expression quantification of TEs. Multi-mapped reads could cause ambiguity when analysing the local effects of TEs on proximal genes as the repeats have many copies on the genome (Goerner-Potvin & Bourque, 2018; Treangen & Salzberg, 2011). To remove multi- and unmapped reads from BAM files, “samtools view -bq 60 -o {out.bam}{input.bam}” command was used. If the user would like to include the multi-mapped reads, they can skip this last command.

Results

Overview of the TEffectR package pipeline

The TEffectR package includes a set of functions (Fig. 1) that allows the identification of significant associations between TEs and nearby genes for any species whose repeat annotation is publicly available at the RepeatMasker website (http://www.repeatmasker.org/genomicDatasets/RMGenomicDatasets.html). Currently, the complete annotation for over 60 species (from primates to nematodes) can be downloaded from this main repository and can be analyzed with our R package. The TEffectR package works initially by manipulating the repeat annotation file and make it ready for downstream analysis. In the following step, our tool takes a raw count matrix of RNA-seq dataset from the user where the first column includes the gene symbols or Ensembl IDs and the other columns contain count values of genes across samples. Then, TEffectR retrieves genomic position of each gene in the respective genome using the given count matrix. Afterwards, based on the user-defined parameters, TEffectR determines all TE species that are located within the upstream regions of each gene individually.

Figure 1 The workflow of TEffectR package.

The package contains six core functions for the identification of the potential links between TEs and nearby genes at genome-wide scale. TEffectR requires two inputs provided by the user: (i) a raw gene count matrix and (ii) genomic alignments of sequencing reads in BAM file format.

The TEffectR package contains a handy function for obtaining sequencing read counts, which are aligned to each TE region from a given list of sorted and indexed BAM files. Additionally, it can calculate the total read counts of each TE associated with a certain gene. In the following step, TEffectR merges all read count values of both genes and TEs into a single count matrix. This count matrix is then filtered, normalized with Trimmed Mean of M-values (TMM) method and transformed for linear modeling using voom() function of the limma package. In the final step, TEffectR fits a linear regression model with customized design matrix for each gene, and it calculates adjusted R-square values and significance of the model, and estimates the model parameters. The users can output the results of all calculations in tab separated values (tsv) file format to assess the contribution of each covariate (e.g., individual repeats) to the model.

Descriptions of the functions in the TEffectR package

The TEffectR package provides six unique functions for predicting the potential influence of TEs on the transcriptional activity of proximal genes in the respective genome:

TEffectR::rm_format: This function takes RepeatMasker annotation file as input and extracts the genomic location of each TE along with repeat class and family information. The output of rm_format() function is used while searching TEs that are located in the upstream region of the genes of interest.

TEffectR::get_intervals: This function is used to retrieve the genomic locations of all genes in a given read count matrix by the user. Row names of the expression matrix must be one of the following: (i) official gene symbol, (ii) Ensembl gene or (iii) transcript ID. The output of this function is utilized while determining distance between genes and TEs.

TEffectR::get_overlaps: Takes the genomic intervals of genes and TEs as input. Besides, the user also requires to provide three additional parameters: (i) the maximum distance allowed between the start sites of genes and TEs, (ii) whether genes and TEs must be located in same strand and (iii) TE family or subfamily name (e.g., SINE, LINE). This function helps to detect TEs that are localized upstream of the genes of interest. The “distance” parameter of this function could be determined by the user based on the interest of the TE localization. The user can either give a positive value, which would take the TEs localized in the upstream region of the gene; or a negative value that would correspond to the downstream of the gene. Moreover, the absolute distance is not limited; however, we used the value of “5000” to test our R package with the TEs located within the 5kb upstream regions of the genes, as previous studies confirmed that TEs located within 5kb upstream of genes provide binding sites for transcription factors and provide a possible link to nearby gene expression (Bourque et al., 2008; Nikitin et al., 2018). However, if the user aims to study long-range effects, then the distance could be set to a higher value.

TEffectR::count_repeats: This function returns a raw count matrix of the total number of reads originated from TE sequences. Only the reads exhibiting 100% overlap with given TE regions are considered and the user needs to specify individual path of each BAM file as input.

TEffectR:: summarize_repeat_counts: Takes the output of count_repeats() function as input. It is used to calculate the total number of sequencing reads derived from each TE that is located upstream of a certain gene.

TEffectR::apply_lm: This core function applies filtering (≥10 reads), TMM normalization, voom transformation and LM to the given raw count expression values, respectively. It takes four arguments: (i) raw gene counts, (ii) raw TE counts, (iii) a data frame containing user-defined covariates (e.g., tissue type, disease state), and (iv) the output of get_overlaps() function. When covariates are determined, one may include all the biological factors to see if they could explain the expression of the gene in conjunction with TE expression. However, one may as well only use the TE expression as the single predictor without the inclusion of further covariates.

The apply_lm() function returns three outputs: (i) a tsv file containing the p-value of each model, significance level of covariates and associated adjusted R squared values. The generated tab delimited file contains the list for the LM results of all genes that have at least one TE within the region of interest as given by this function. (ii) another tsv file containing log2(CPM) values of genes and TEs included in LM, and (iii) a group of diagnostic plots for each significant model (p < 0.05).

Table 1 Examples of significant associations of LINE, SINE, LTR and DNA transposons with genes that were previously linked to breast cancer as TEffectR outputs along with multiple covariates.

Expression levels of TEs within the upstream 5 kb regions of the given genes and other covariates such as the tissue type (healthy vs. tumor) or patient number were included in the linear regression model. The p-value of the model indicates the significance of the linear model. P-values for each covariate indicate whether these factors have significant associations with the expression of the given gene. Adjusted r-square score indicates the precision of the model with significant covariate associations in terms of predicting the expression of the gene. For example, an adjusted R square of 0.8422 indicates that the linear model could explain 84.22% of the gene’s expression.

Link to breast cancer	Gene name	TE name	r squared	Adjusted r-squared	Model p-value	Individual p-values	
Biomarkera	KRT8 (CK8)	L2c (LINE)	0.8532	0.8422	1.026E–16	L2c: 1.356E–13
Tissue type: 0.0332
Patient: 0.7974	
Prognosisb	SLC39A6 (LIV-1)	L2b
(LINE)	0.7231	0.7023	3.100E–11	L2b: 7.536E–08
Tissue type: 0.0013
Patient: 0.1024	
Molecular pathogenesisc	SAFB	L1MB7 (LINE)	0.5131	0.4766	2.107E–06	L1MB7: 1.114E–07
Tissue type: 0.6112
Patient: 0.1394	
Susceptibilitydand prognosise	CHEK2	AluJb, AluSx AluS
(SINE)	0.6362	0.5883	1.645E–07	AluJb: 0.0433
AluSx: 0.0426
AluSz: 0.0005
Tissue type: 0.0023
Patient: 0.0033	
Susceptibilityfand prognosisg	FEN1	MIR3
(SINE)	0.5545	0.5211	3.703E–07	MIR3: 2.572E–06
Tissue type: 0.0122
Patient: 0.3886	
Molecular genetics and pathogenesish	CENPL	AluSx3, AluY
(SINE)	0.5489	0.5027	2.118E–06	AluSx3: 0.0007
AluY: 0.2000
Tissue type: 0.0066
Patient: 0.2467	
Prognosisi	MCM4	MLT1D
(LTR)	0.5733	0.5413	1.587E–07	MLT1D: 0.0012
Tissue type: 1.544E–06
Patient: 0.1674	
Susceptibilityj	RMND1	LTR5_Hs
(LTR)	0.4318	0.3892	4.279E–05	LTR5_Hs: 1.782E–05
Tissue type: 0.4280
Patient: 0.1193	
Molecular pathogenesis and prognosisk	CPNE3	MLT1H2
(LTR)	0.3910	0.3453	0.0002	MLT1H2: 0.0002
Tissue type: 0.0055
Patient: 0.9407	
Biomarker and prognosisl	HLA-DPB1	hAT-1_Mam
(DNA)	0.8318	0.8192	1.548E–15	hAT-1_Mam: 1.092E–14
Tissue type: 0.5467
Patient: 0.2850	
Molecular pathogenesismand biomarkern	HSPB2 (HSP27)	MER5B
(DNA)	0.7756	0.7587	4.791E–13	MER5B: 8.050E–07
Tissue type: 0.5756
Patient: 0.1733	
Molecular pathogenesiso	PARP9	MER5B
(DNA)	0.5929	0.5624	6.287E–08	MER5B: 1.141E–06
Tissue type: 0.0054
Patient: 0.5464	
Notes.

a Heo et al. (2013).

b Kasper et al. (2005).

c Hammerich-Hille et al. (2010).

d Nagel et al. (2012)

e Li et al. (2014); Li, Liang & Zhang (2014)

f Chung et al. (2015).

g Abdel-Fatah et al. (2014).

h Tishchenko et al. (2016).

i Kwok et al. (2015).

j Dunning et al. (2016).

k Heinrich et al. (2010).

l Forero et al. (2016).

m Wei et al. (2011).

n Storm et al. (1995).

o Tang et al. (2018).

A case example of TEffectR analysis using the RNA-seq data obtained from healthy and tumor tissues of ER+/HER2- breast cancer patients

Breast cancer pathogenesis was associated with genomic instability (Kwei et al., 2010), which often presents itself with the aberrant expression of TEs (Aguilera & Garcia-Muse, 2013; Burns, 2017). TEs including LINE, SINE and LTR elements were already shown to be dysregulated in this disease (Yandım & Karakülah, 2019a; Bakshi et al., 2016; Bratthauer, Cardiff & Fanning, 1994; Johanning et al., 2017); with little or no information on the subtypes of these repeats. Also, there is a paucity of information on the impact of such dysregulatory events on the expressions of genes. To demonstrate the usage of the TEffectR package on a real case example, we ran the TEffectR package on the transcriptome RNA-seq data set of ER+/HER2- breast cancer patients. Table 1 summarizes a group of significant genes involved in breast cancer pathogenesis, diagnosis and prognosis (Abdel-Fatah et al., 2014; Chung et al., 2015; Dunning et al., 2016; Forero et al., 2016; Hammerich-Hille et al., 2010; Heinrich et al., 2010; Heo et al., 2013; Kasper et al., 2005; Kwok et al., 2015; Li et al., 2014; Storm et al., 1995; Tang et al., 2018; Tishchenko et al., 2016; Wei et al., 2011), where TEffectR presented a linear regression model that shows the associations between the expressions of these genes and that of the uniquely mapped TE sequences located within their upstream 5 kb flanking regions (Fig. 2). These genes were only given to present a contextual example and were selected based on breast cancer literature from the tab-delimited file that contains all genes with at least one TE in their upstream regions. The LM could explain the effect of these TEs on the variation in gene expression along with other covariates such as the type of tissue (i.e., healthy or tumor) or the individual patients. For example, two of the dependent variables; the LINE element “L2b” and “Tissue type” (p = 0.0013) could statistically significantly predict 70.30% of the expression of the “SLC39A6 (LIV-1)” gene whereas the covariate “patient” (p = 0.1024) could not explain the variation in this particular gene’s expression in this statistically significant model (p < 0.001). On the other hand, even though the expression of LTR5_Hs could predict 38.92% of “RMND1” expression statistically significantly (p < 0.001), neither the type of tissue (healthy or tumor; p = 0.4280) nor the individual patient (p = 0.1193) could explain the expression of this gene. From the perspective of a molecular biologist, these results may imply that SLC39A6 gene could potentially be involved in the tumorigenesis of the breast whereas this was not the case for RMND1, and the relevant repeat motif upstream of both genes could be suitable for further experimental investigation in terms of its potential to influence the expression of the proximal gene. These results may have implications on the biological roles of TEs (e.g., L2b) on breast cancer-related gene expressions (e.g., SLC39A6) and could indicate their potential roles in the carcinogenesis of the breast where the tissue type (healthy vs. tumor) p-value of the LM result is less than a significant threshold (i.e., p < 0.05).

Figure 2 Scatter plots that demonstrate the correlations of normalized read counts of genes given in Table 1 with the normalized read counts of TEs present in their upstream 5-kb regions.

(CPM: counts per million).

Discussion

Repetitive DNA and its regulatory effects on chromatin environment and gene expression have been recognized well since the early years that follow the discovery of chromatin modifications (He et al., 2019; Huda et al., 2009; Martens et al., 2005). Dynamic expression patterns of TEs during distinct stages of human embryonic development (Garcia-Perez, Widmann & Adams, 2016; Grow et al., 2015; Yandim & Karakulah, 2019b), where the whole genome is tightly regulated in a highly orchestrated manner, and the power of TEs to modify gene expression patterns by various routes (Garcia-Perez, Widmann & Adams, 2016), highlight the importance of studying the links of TE expression with proximal gene transcription. TEffectR does not only provide a linear regression model between the expression of TEs and a gene in a given genomic location, but it also presents a platform to make this information traverse through biological contexts such as cancer, treatment, age, etc. The option of adding a desired number of covariates along with TEs that are present in a desired distance interval from a given gene allows one to study the associations between TEs and genes along with multiple factors. Substantial studies suggest that some human gene promoters are derived from TEs (Cohen, Lock & Mager, 2009) and that some TEs could act as distal enhancers (Kunarso et al., 2010). Still, it should be noted that significant associations documented via TEffectR do not necessarily mean that a given TE indeed has an influential effect on the transcription of the proximal gene. Conversely, transcriptional activation of a given gene could also influence the expression of the nearby TE, and the TE might not have an effect on gene expression at all. This is why functional experiments should always be performed to clearly answer crucial biological questions regarding this matter, where TEffectR acts as a useful guideline to point out significant associations.

Conclusion

The highly complex interactions among the regulatory networks of the genome are at the center of attention of many areas of molecular biology, developmental biology and epigenetics. Here, we present TEffectR, an R package, which elaborately dissects the associations between the expressions of genes and the transposable elements nearby them in a unified linear regression model. The inclusion of a desired number of factors as covariates allows a biologist to study such associations in a broader context.

Additional Information and Declarations

Competing Interests

Author Contributions

Data Availability

Gökhan Karakülah and Aslı Suner are Academic Editors for PeerJ.

Gökhan Karakülah conceived and designed the experiments, performed the experiments, analyzed the data, contributed reagents/materials/analysis tools, prepared figures and/or tables, authored or reviewed drafts of the paper, approved the final draft.

Nazmiye Arslan conceived and designed the experiments, performed the experiments, contributed reagents/materials/analysis tools, authored or reviewed drafts of the paper, approved the final draft.

Cihangir Yandım conceived and designed the experiments, performed the experiments, analyzed the data, prepared figures and/or tables, authored or reviewed drafts of the paper, approved the final draft.

Aslı Suner conceived and designed the experiments, analyzed the data, contributed reagents/materials/analysis tools, prepared figures and/or tables, authored or reviewed drafts of the paper, approved the final draft.

The following information was supplied regarding data availability:

The analysis pipeline is available at https://github.com/karakulahg/TEffectR.

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
