# Peer review of "TEffectR: an R package for studying the potential effects of transposable elements on gene expression with linear regression model"

_PeerJ, doi:10.7717/peerj.8192_

## Round 0.1 · original submission · Major Revisions

The manuscript has been assessed by two external reviewers who agree on the fact that the manuscript is nicely written but there are several points that need to be addressed in the implementation and analyses, along with documentation. We would be glad to consider a substantial revision of your work, where the reviewers' comments will be carefully addressed one by one.

·

Basic reporting

The article is well written, with the standard sections.

The code to use their algorithms is provided through github, which makes it easy to use, although there are shortcomings explained in "Experimental design".

Table 1 is missing some text in the caption in the pdf, it is fine in its own document.

A great job has been done with the bibliography, with a good search of genes related to breast cancer.

Experimental design

The expression "nearby genes" is used in several locations to decide which genes are affected by a TE. but it is only defined on line 234, and in several captions.
It can be modified by the user in the algorithm, but the maximum distance is a basic parameter in this article, so it should merit some reasoning as to why use 5kb by default, or why use other distances.

No justification is provided for discarding the multimapped reads (line 158), it should be motivated. Also the minimum amount of reads to provide a reasonable result should be indicated, because due to the discarding of the multimapped reads it seems to me there is going to be few reads counted as TE.

The use of covariants dilutes the effect of the TE in the lm, using other variables to fine tune the lm. The only justification I have found to use covariants is that the results are better, (line 277).

Validity of the findings

Only the effect on known genes is looked up, so it seems the algorithm is to first look for important genes in the literature, then analyze nearby TEs, instead of first looking at the TEs and then providing the genes that matches the best.

Several times it says "health vs tumor" tissues, but it seems it isn't covered in the results, it only uses the "health vs tumor" as another covariant to provide a better lm.

Additional comments

No comment

Reviewer 2 ·

Basic reporting

The authors investigated the effect of transposable elements on gene expression with regression model, which is a very relevant topic to the field at the moment. They have based theirs results on the correlation between the expression of TEs and a gene in a give genomic location developing a tools to calculate the parameters of correlation. However, this correlation not show the effect the expression of TEs relating to his adjacent genes.

Experimental design

The authors use a mapping bam file to evaluate the expression TEs, nevertheless they describe perfunctorily which are the parameters of the alignment. The method should be described with sufficient information. On the other hand, even though the multi mapping not be as precise as single mapping reads, you can get a better determination of which loci and types of elements are unregulated in your samples whether you evaluate the multimapping reads. I think that the multi-mapping reads could to be analyzed with Squire https://doi.org/10.1093/nar/gky1301 or with TEtranscripts DOI: 10.1093/bioinformatics/btv422-

The authors use a Repeat Masker TEs to localitation TEs on genome, however this database contain both activa a inactive TEs so it is very difficult to ensure that the expression TE has effect on the expression genes.

On this sense, in the title, the autor focus us on the effect the expression TE on near genes, however in the text they say: ‘Still, it should be …… proximed genes’, so the title do not reflect the result of this work', a title overambitious.

Validity of the findings

If the author used a dataset normal and tumor tissue, why they focus only a few genes? Why they only investigated genes at 5 kb upstream? why not other distances? In the heathy tissue, which is the expression this TEs?

It would be helpful if the authors could something comment on methylation in these tumors and whether or not it is likely contributing to the expression patterns, this would add some rigor to the study and further support the results without needing to perform additional experiments. It would further be helpful if the authors would state some how that if the methylation differences were the reason, one would expect global differences for different classes of elements or theirs near genes.

---

## Round 0.2 · accepted · Accept

I am glad to accept this manuscript for publication.

·

Basic reporting

It continues to be well written, with the standard sections.

Experimental design

All questions presented in the first review have been answered correctly.

Validity of the findings

All questions have been answered correctly. The article introduces an improvement of existing algorithms and programs.

Reviewer 2 ·

Basic reporting

Overall, the submitted paper has been improved regarding to the older version, is well-written with sufficient intro about the background of the TE and its potential relationship with cancer. Also results and discussion has been improved, just as the tittle.
The authors has answered all my concerns, so, I recommend this paper to be published in PeerJ .

Experimental design

No comment

Validity of the findings

No comment

Additional comments

No comment